# Interaction between bacterial microbiota and nematode parasite communities in sheep's gastrointestinal tract

**Laura Mate[1], Luis Ignacio Alvarez[1], Mercedes Lloberas[2], Fernanda Imperiale[1], Carlos Edmundo Lanusse[1], Juan Pedro Liron[1]\***

1 Laboratorio de Farmacología, Centro de Investigación Veterinaria de Tandil (CIVETAN), UNCPBA-CICPBA-CONICET, Facultad de Ciencias Veterinarias, UNCPBA, Tandil, Argentina, 2 Instituto de Innovación para la Producción Agropecuaria y el Desarrollo Sostenible (IPADS Balcarce) EEA-INTA, Balcarce, Argentina

\* juanpedroliron@gmail.com

**Data Availability Statement:** All relevant data are within the manuscript and its Supporting Information files.

## Abstract

The economic impact of gastrointestinal (GI) nematode infections on livestock production is well documented worldwide. Increasing evidence supports the hypothesis that parasite colonization induces significant changes in the GI tract environment and, therefore, in the landscape where the microbiota and parasites occur. Understanding the interactions between bacterial and parasite populations in the digestive tract of livestock may be useful to design parasite control strategies based on microbiota modification. The aims of this work were to investigate the impact of the oxytetracycline-mediated manipulation of the gut microbial community on the composition of GI nematode populations in naturally infected sheep and to explore changes in the GI microbial communities after nematode population treatment with the anthelmintic compound monepantel. Extensive manipulation of the GI microbiota with a therapeutic dose of the long-acting oxytetracycline formulation did not induce significant changes in the GI nematode burden. The gut microbiota of treated animals returned to control levels 17 days after treatment, suggesting strong resilience of the sheep microbial community to antibiotic-mediated microbiota perturbation. A significant decrease of the bacterial *Mycoplasmataceae* family ($Log_2FC = -4$, $P_{adj} = 0.001$) and a marked increase of the *Methanobacteriaceae* family ($Log_2FC = 2.9$, $P_{adj} = 0.018$) were observed in the abomasum of sheep receiving the monepantel treatment. While a comprehensive evaluation of the interactions among GI mycoplasma, methanobacteria and nematode populations deserves further assessment, the bacteria-nematode population interactions should be included in future control programs in livestock production. Understanding how bacteria and parasites may influence each other in the GI tract environment may substantially contribute to the knowledge of the role of microbiota composition in nematode parasite establishment and the role of the parasites in the microbiota composition.

**Funding:** This work was supported by CONICET PIP 11220100102665CO and the Agencia Nacional de Promoción Científica y Tecnológica PICT 2020-0343.

**Competing interests:** The authors have declared that no competing interests exist.

## Introduction

Gastrointestinal (GI) nematode infections in livestock continue to cause substantial economic impacts worldwide due to animal production losses. Over the last 50 years, GI nematode control in livestock has heavily relied on the repeated treatment with synthetic drugs [1]. The intensive and often indiscriminate use of these drugs has led to selection of resistant strains of numerous helminth species of veterinary importance. Therefore, anthelmintics become less effective or completely ineffective over time [2]. Thus, alternative parasite control strategies should be developed to extend the lifespan of available anthelmintic drugs.

In mammals, many studies have shown the association between GI helminth infections and both the composition and functionality of the host's gut microbiota [3–5]. The gut microbiota is a complex and diverse community of microorganisms residing in the GI tract, including bacteria, viruses, fungi, and other microorganisms that coexist in a synergistic balance [6]. These microbes play crucial roles in maintaining the host health, including digestion, nutrient absorption, immune system regulation; in humans, they even influence certain aspects of neurological disorders [7]. A healthy intestinal microbial community is diverse, stable, resistant to changes, and resilient [8]. The imbalance or disruption in the natural composition of microbial communities (dysbiosis) within the GI microbiota is associated with the emergence of various acute and chronic inflammatory diseases, GI issues, metabolic disorders and skin conditions [9].

Increasing evidence supports the hypothesis that parasite colonisation of the GI tract significantly changes the physiological characteristics of the gut environment and hence the landscape in which the microbiota and parasites reside. The presence of certain parasites in the GI tract of different mammal species induces changes in intestinal mucus production, composition and structure; decreases bacteria attachment to the gut epithelium; alters food availability for bacteria; and increases the microbes that use mucus as a carbon and energy source [4, 10, 11]. Recently, Cortés et al. (2020) [12] described that *Teladorsagia circumcincta* infection in sheep leads to an expansion of potentially pro-inflammatory gut microbial species and abomasal T cells. On the other hand, the composition of bacterial microbiota in piglets and rats induced changes in the expulsion rate of parasites by alterations in the mucus synthesis [13, 14]. Moreover, both microbes and helminthic parasites shape the immune response of the gut by changing the expression of Toll-like receptors in macaques and cultured splenocytes [15, 16]. In rodents, helminth body secretions have antibacterial properties [17] and microbiota-derived metabolites hinder parasite infection by triggering an inflammasome response [18].

While challenging, understanding how GI parasites and microbes influence each other could promote the development of novel microbiome-targeting approaches and other bacteria-based strategies for parasite control. For instance, the intervention of the gut microbial communities, e.g., via the administration of probiotics and prebiotics, and the use of bioengineered bacteria and nutritional supplements, has been postulated as an alternative strategy for parasite control and management [3, 5, 19, 20]. Despite the importance of GI nematode parasitism in domestic ruminants, little is known about the nature and strength of the interactions among GI helminths and microbiota. Thus, the aims of the current work were to investigate, under *in vivo* conditions, whether antibacterial-mediated modifications in sheep GI bacterial communities influence parasitic nematode populations, and how anthelmintic-mediated changes in nematode burdens affect the microbial community in the GI tract.

## Materials and methods

### Animals and experimental design

To study how manipulation in gut sheep microbial and nematode communities influence each other, we used a long-acting formulation of the broad-spectrum antimicrobial drug

oxytetracycline (OTC-LA) and the anthelmintic compound monepantel (MNP). For this purpose, 30 Corriedale lambs (24.5 ± 4.37 kg) naturally infected with GI nematodes were selected. Infection was confirmed by parasitological examination of individual faecal samples before the beginning of the study. The animals were in a paddock on an experimental sheep farm (Reserva 8, Instituto Nacional de Tecnología Agropecuaria, Balcarce, Argentina) with records of resistance to IVM and benzimidazole drugs [21–23]. The animals grazed on implanted pasture of *Trifolium pratense*, *Trifolium repens* and *Agropirum*. They had unrestricted access to water during the experiment days.

At day 0 faecal samples were taken from the 30 lambs to perform the faecal egg count. Fifteen lambs were selected and assigned to three experimental groups (Control, MNP and OTC-LA) of five lambs each. The groups were balanced in terms of sex and parasite burden. Animals belonging to the MNP group were treated with MNP (2.5 mg/kg orally, Zolvix, Elanco, Argentina) at days 0, 7 and 14; animals belonging to the OTC-LA group were treated with OTC-LA (20 mg/kg subcutaneously, Terramicina LA, Zoetis, Argentina) at day 0; animals in the Control group were not treated. The antiparasitic treatment consisted of three doses to prevent animal reinfection with larvae present in the pasture. After a period of microbiota stabilization, at day 17 the animals were sacrificed by captive bolt gun and rapidly exsanguinated. Efforts were made to minimize suffering.

Content samples from the abomasum, and the small and large intestine, and faecal samples were taken for microbiome and parasitological analyses. The Ethics Committee (Animal Welfare Policy, act (2/2013) of the Faculty of Veterinary Sciences, Universidad Nacional del Centro de la Provincia de Buenos Aires (UNCPBA), Tandil, Argentina (http://www.vet.unicen.edu.ar) approved the animal procedures and management protocols. Adult parasites were counted in content samples from the abomasum, and the small and large intestine. Intestinal content collected for microbiome analyses was stored in sterile tubes and immediately transported to the laboratory and stored at -72°C until DNA extraction. Faecal samples were cultured to obtain L3 for nemabiome analysis.

## Parasitological analyses

Individual faecal egg counts were performed using the McMaster technique modified by Roberts and O'Sullivan (1949). Adult parasites collected from the abomasum, and the small and large intestine were counted and identified throughout the methodology applied to perform the controlled efficacy test (CET), following the World Association for the Advancement of Veterinary Parasitology (WAAVP) guidelines [24]. Moreover, a pool of 2000 $L_3$ obtained by culturing a pool of faeces from each experimental group at day 17 was used to perform ITS-2 gene deep amplicon sequencing (nemabiome) at the Department of Comparative Biology and Experimental Medicine, Calgary, Canada [25]. This method was performed to assign the species to each parasite genus recovered from each intestinal segment. Adult parasite counts were analysed by a Wald test. A false discovery rate (FDR) adjusted p-value $< 0.05$ was considered statistically significant.

## Bacterial DNA extraction and 16S amplicon metagenomic sequencing

Bacterial DNA was extracted from 0.20 g of abomasum, and small and large intestine content using the QIAamp PowerFecal Pro DNA kit (Qiagen, California, USA). The quality and quantity of DNA were measured on a Nanodrop. DNA samples (100 μl) were stored at -72°C until Novogene Shipment for 16S amplicon metagenomic sequencing. High-throughput sequencing of the V3-V4 hypervariable region of the bacterial 16S rRNA gene was amplified with forward primer 341F (CCTAYGGGRBGCASCAG) and reverse primer 806R

(GGACTACNNGGGTATCTAAT), with sequencing adaptors at the 5´ end. The PCR product was purified and concentration and quality were determined for each amplicon using Qubit™ fluorometer and Agilent Bioanalyzer 2100. Barcoded amplicons were sequenced with the Illumina NovaSeq 6000, HiSeq (Novogene) platform.

### Microbiome analyses

The FASTQ-files were imported to QIIME2; the DADA2 plugin was used to denoise and quality filter reads. A naïve Bayes classifier was trained against the SILVA v138 database to include only the V3-V4 region and assign taxonomy to the sequences [26]. The OTU table, taxonomy, metadata, and phylogenetic tree were imported into the R package Phyloseq [27]. Sequences derived from chloroplasts, mitochondria or eukaryote were removed. Library rarefaction was applied to calculate alpha and beta diversities among samples at the even depth of 90% of the sample with the lowest number of reads (50,495 for the abomasum and 83,772 for the large intestine). Alpha diversity was estimated with the Shannon index and number of observed OTUs. The non-parametric Kolmogorov-Smirnov test was used for two-group comparisons (Microbiome R package; [28]). Beta diversity was calculated using Bray-Curtis and unweighted UniFrac distances. A principal coordinate analysis (PCoA) based on the Bray-Curtis and unweighted UniFrac distances was conducted. They were plotted with the first two axes that captured the greatest variance in the data. To test for significant differences between the control and the treatment groups, a PERMANOVA was performed using the Adonis function from the Vegan Package [29]. Differential relative abundance of OTUs at the phylum, class, order, and family levels between sample groups was calculated using DESeq2 [30]; only bacterial families with a relative abundance higher than 1% were considered for discussion.

## Results

### Parasitological analyses

Only *Oesophagostomum radiatum* and *Trichuris ovis* were recovered in the MNP-treated animals; they were present in the large intestine of treated sheep in low numbers. The adult parasite counts in the content of the abomasum, and the small and large intestine obtained in Control and OTC-LA groups are shown in Fig 1. ITS-2 gene deep amplicon sequencing of a pool of 2000 $L_3$, obtained by culturing a pool of faeces from each experimental group at day 17, allowed us to assign the species to the parasites infecting sheep. *Trichostrongylus* (both *colubriformis* and *axei*) was the most prevalent GI nematode, followed by *Cooperia oncophora* and *Haemonchus contortus*. The Wald test did not show significant differences in parasite content between Control and OTC-LA groups (Table 1).

### Microbiome analyses

DNA with the quality and amount required for metagenomic analysis was obtained from the abomasum and the large intestine content. By contrast, after two unsuccessful attempts, only in 6 of 15 small intestine samples was it possible to obtain the DNA required for metagenomic analysis. For this reason, samples from the small intestine were not included for metagenomic sequencing.

After quality control and removal of chimeras, the library size varied from 59320 to 125878, with a mean of 108593 reads per sample. After the filtering process all the samples met the requirements for subsequent bioinformatics analyses (S1 Table).

**Abomasum.** A total of 1716 bacterial operational taxonomic units (OTUs) were found. At the taxonomic level, 18 phyla, 28 classes, 50 orders and 83 families of bacteria (S2 Table) were

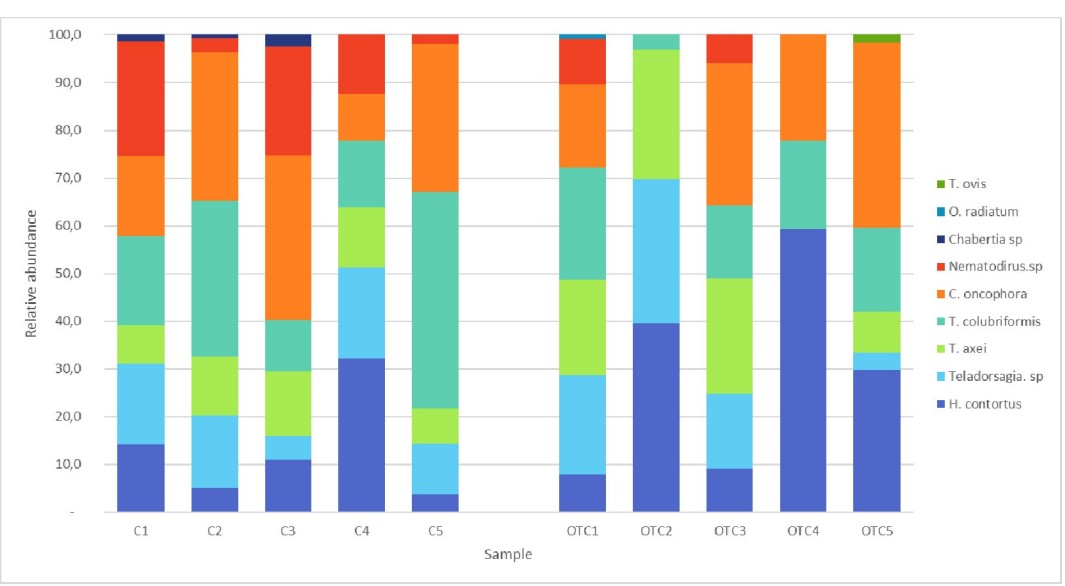

**Fig 1. Nematode abundance in the gastrointestinal tract of untreated (Control) sheep and sheep treated with long acting oxytetracycline (OTC-LA).** The relative proportion of each parasite species present in each animal belonging to Control (C1-C5) and OTC-LA (OTC1-OTC5) groups, as determined by nemabiome metabarcoding (ITS-2 rDNA deep-amplicon sequencing), is indicated.

identified. The most abundant phyla in the abomasum microbiome of sheep were *Firmicutes* (45.28%), *Bacteroidota* (37.94%) and *Actinobacteriota* (4.91%), followed by *Fibrobacterota* (3.86%), *Spirochaetota* (2.57%), *Proteobacteria* (1.56%) and *Euryarchaeota* (1.39%). The remaining detected phyla had a relative abundance lower than 1%. At the family level, the highest abundances corresponded to *Prevotellaceae* (13.43%), *Rikenellaceae* (9.81%) and *Lachnospiraceae* (8.63%).

The relative abundance of OTUs at the phylum, class, order, and family levels was analysed (S2 Table). Thus, in dewormed animals, the *Mycoplasmataceae* family, class *Bacilli*, phylum *Firmicutes* ($Log_2FC$ = -4, $P_{adj}$ = 0.001) decreased significantly (Fig 2), whereas the *Methanobacteriaceae* family, class *Methanobacteria*, phylum *Euryarchaeota* ($Log_2FC$ = 2.9, $P_{adj}$ = 0.018) (Fig 2) and *Enterococcaceae*, class *Bacilli*, phylum *Firmicutes* ($Log_2FC$ = 5.0, $P_{adj}$ = 0.001) increased. Regarding the relative abundance of OTUs in the abomasum of animals receiving the antibiotic-treatment, a significant decrease was detected in the *Absconditabacteriales*

**Table 1. Parasite counts in the abomasum, and the small and large intestine of sheep belonging to Control and OTC-LA groups.**

| Location | Parasite | Mean | | Wald test |
|---|---|---|---|---|
| | | Control | OTC-LA | padj |
| | *H. contortus* | 3000 | 2560 | 0.47 |
| Abomasum | *T. circumcincta* | 2720 | 1800 | 0.70 |
| | *Trichostrongylus axei* | 1980 | 2280 | 0.53 |
| | *Trichostrongylus colubriformis* | 4260 | 1980 | 0.47 |
| Small intestine | *C. oncophora* | 4000 | 2920 | 0.53 |
| | *Nematodirus sp* | 2460 | 560 | 0.47 |
| | *Chabertia sp* | 140 | 0 | 0.53 |
| Large Intestine | *O. radiatum* | 0 | 20 | - |
| | *Trichuris ovis* | 0 | 20 | - |

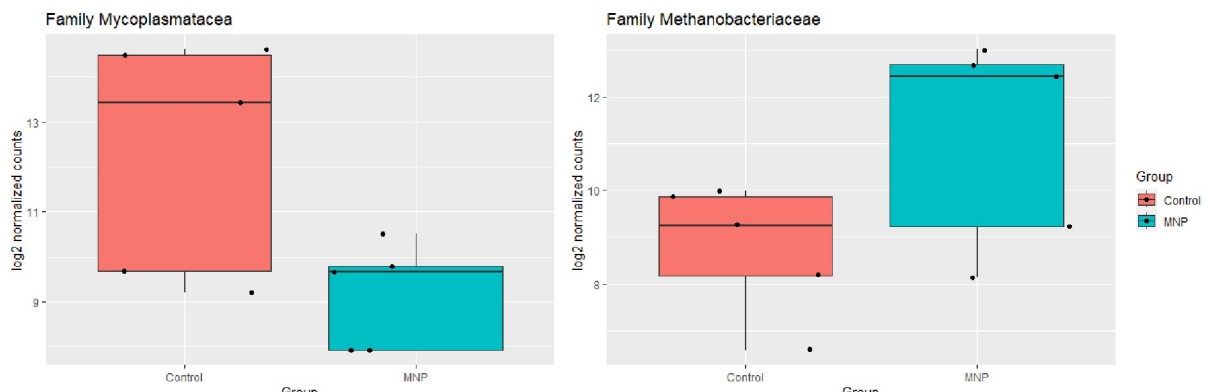

**Fig 2. Differential abundance of *Mycoplasmatacea* and *Methanobacteriaceae* families in the abomasum of Control and monepantel (MNP)-treated sheep.** The boxplot shows the normalized counts of *Mycoplasmatacea* and *Methanobacteriaceae* families in the abomasum of Control and monepantel (MNP) groups. The horizontal dashed lines represent the median values for each corresponding value. Dots are observations of each animal.

family, class *Gracillibacteria*, phylum *Patescibacteria* ($Log_2FC$ = -2.1, $P_{adj}$ = 0.009), but an increase in the family *Clostridia Vadim BB60*, class *Clostridia*, phylum *Firmicutes* ($Log_2FC$ = 1.3, $P_{adj}$ = 0.041). However, both *Absconditabacteriales* and *Clostridia Vadim BB60* families had a very low abundance (less than 0.2%) in the abomasum of the studied animals.

Alpha diversity, measured by the number of observed OTUs and the Shannon index (Fig 3A), varied between abomasum samples, ranging from 962 to 1716 and from 5.10 to 6.53, respectively. The Kolmogorov–Smirnov test did not reveal differences in Shannon diversity between groups, either in Control *vs.* MNP (Padj = 0.36) or in Control *vs.* OTC-LA (Padj = 0.87). The variation in community composition between different sample distances among groups was calculated using Bray-Curtis and unweighted UniFrac β dissimilarity distances. The PCoA plots (Fig 4A, 4B) show that samples from the Control and OTC-LA groups had a more similar microbiota structure than three samples from the MNP group (Bray-Curtis PERMANOVA, p = 0.021, Unifrac PERMANOVA p = 0.019).

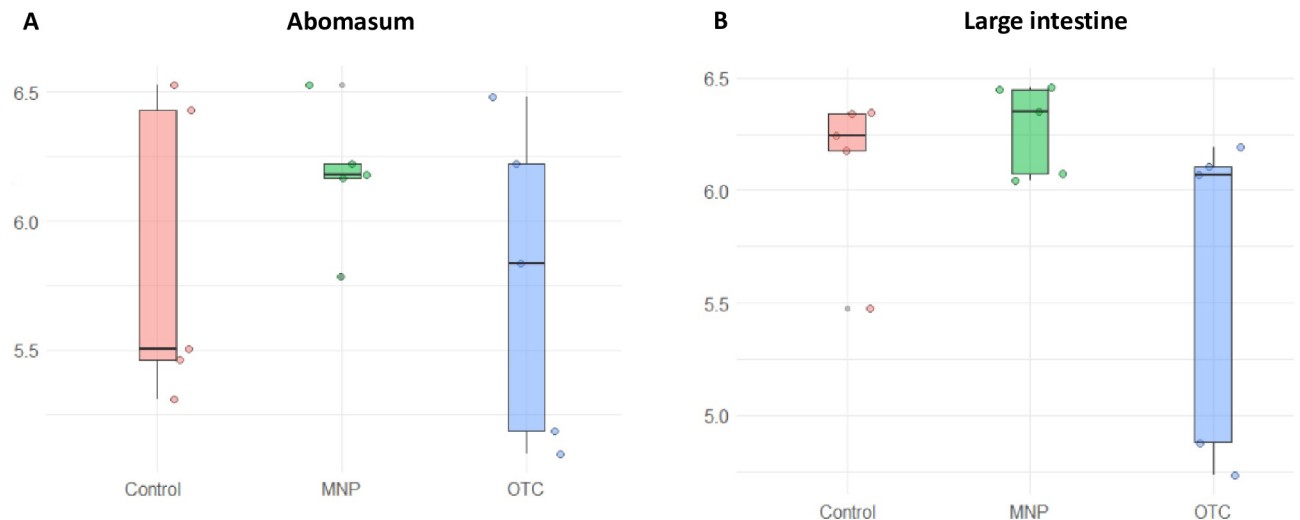

**Fig 3. Estimates of microbiota richness in sheep from Control, monepantel (MNP) and long-acting oxytetracycline (OTC-LA) groups.** The boxplot shows the observed values of the Shannon Index for the abomasum (**A**) and the large intestine (**B**) across 15 animals from Control, MNP and OTC-LA groups. The horizontal dashed lines represent the median values for each corresponding microbiota. Dots are observations of each animal.

**Large intestine.** A total of 1474 bacterial operational taxonomic units (OTUs) were found in the large intestine. In addition, 16 phyla, 22 classes, 43 orders and 75 families of bacteria were identified (S3 Table). The most abundant phyla in the large intestine microbiome of sheep were *Firmicutes* (50.38%), *Bacteroidota* (35.51%) and *Verrucomicrobiiota* (4.95%), followed by *Campilobacterota* (2.55%), *Proteobacteria* (1.59%) and *Actinobacteriota* (1.03%). The remaining phyla detected were characterised by a relative abundance lower than 1%. At the family level, the highest abundances were recorded in *Oscillospiraceae* (14.17%), *Lachnospiraceae* (11.21%) and *Rikenellaceae* (9.48%). The number of observed OTUs and the Shannon index (Fig 3B) in samples from the large intestine range from 690 to 1474 and from 4.73 to 6.46, respectively. The Kolmogorov–Smirnov test did not show differences in Shannon diversity between groups either in Control *vs*. MNP (Padj = 0.357) or in Control *vs*. OTC-LA (Padj = 0.357).

The comparison of the relative abundance of OTUs at the phylum, class, order, and family levels showed that, in the large intestine, 17 days after treatment there were no statistical differences between control animals and animals receiving anthelmintic or antimicrobial drugs, except for the poorly represented family *Paludibacteraceae*, class *Bacteroidia*, phylum *Bacteroidota*, which was higher in animals receiving antimicrobial treatment ($Log_2FC$ = 4.0, $P_{adj}$ = 0.026) (Supplemental material). However, the relative abundance of this family was lower than 1%.

The PCoA plots (Fig 4C and 4D) show that samples from the control and MNP groups had a more similar microbiota structure than three samples from the OTC-LA group (Bray-Curtis PERMANOVA, p = 0.008, Unifrac PERMANOVA p = 0.013).

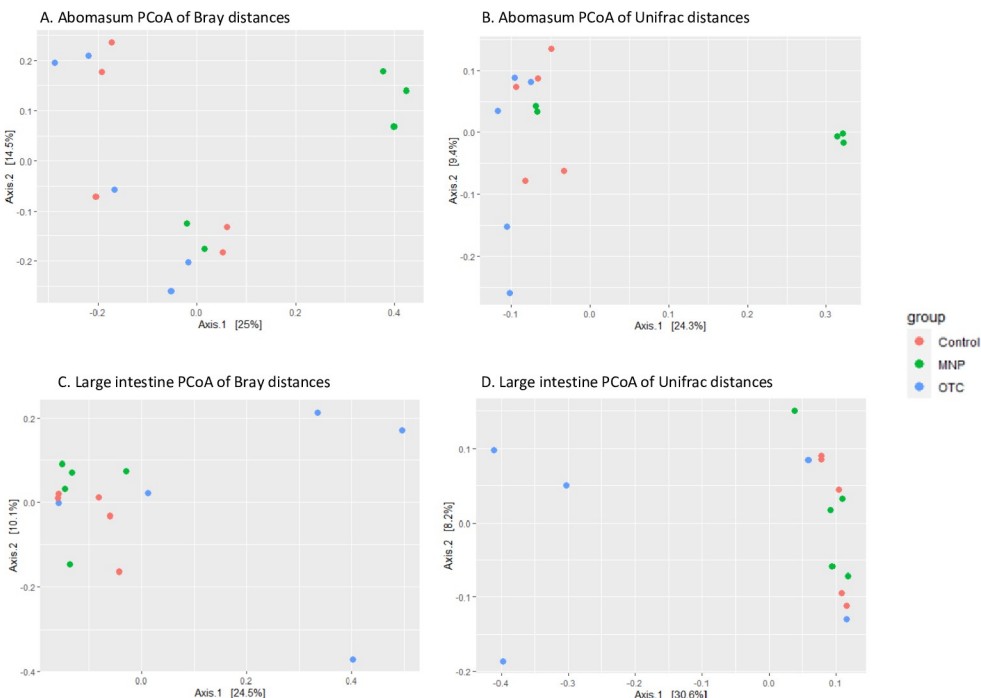

**Fig 4. Principal coordinate plots representing beta diversity on samples.** The plots show comparison of microbiota diversity among monepantel (MNP), long acting oxytetracycline (OTC-LA) and Control groups. (A) Principal coordinate analysis (PCoA) of Bray-Curtis distances for abomasum. (B) PCoA of unweighted UniFrac distances for abomasum. (C) PCoA of Bray-Curtis distances for large intestine. (D) PCoA of unweighted UniFrac distances for large intestine. Sheep from each group are identified with colored dots.

## Discussion

The potential interactions between bacterial microbiota and helminth parasites in the GI tract of livestock species is an important area of research, given the economic losses in production, morbidity, and mortality associated with helminth infections, alongside the escalating threat of resistance to anthelmintic drugs. In this line, a new approach based on the manipulation of symbiotic microbial communities has been suggested for the control of parasitic diseases [31]. To examine how alteration of one community of organisms may affect the other, experiments manipulating sheep intestinal microbiota and nematode community were performed using monepantel and long-acting oxytetracycline. This study was conducted *in vivo*, using naturally infected sheep, to have a good representation of a "real-world" field situation. The antimicrobial and anthelmintic treatments were administered at label doses to ensure that drug concentrations that reach the GI tract are those to which microbes and parasites are commonly exposed after regular drug treatments. Trials involving domestic animals are useful for the collection of data on helminth-microbiota interactions in an ecological context, and findings can be extrapolated from controlled laboratory environments to field settings [4, 32].

In this work, MNP was used to ensure the removal of most parasitic nematodes infecting sheep with the aim to examine how the alteration of this parasite population affects the GI tract microbiota. *Nematodirus* was identified only by faecal egg count because the eggs of this genus do not develop to the larval stage under standard coproculture conditions and, therefore, ITS-2 sequencing cannot be performed. Only *Oesophagostomum radiatum* and *Trichuris ovis* were recovered in the MNP-treated animals; they were present in the large intestine of treated sheep in low numbers. Variable results of the efficacy of MNP against *O. venulosum* and *T. ovis* have been reported [33, 34]. A possible explanation to this variation is that the drug concentrations are lower in the mucosal lining of the large intestine than in that of the small intestine of the treated sheep [35]; however, a reduced sensitivity of these parasites to MNP cannot be ruled out.

Relatively little is known about how parasite burden may affect the host gut microbiota. For example, infection with the GI nematode *Teladorsagia circumcincta* increased the potentially pro-inflammatory gut microbial species in sheep [12]. Mamun et al. (2020) [36] reported significantly more Firmicutes and fewer Bacteroidetes in faeces of sheep with low burden than in sheep with high burden. In helminth-infected humans, albendazole administration produced changes in *Clostridiales* and *Bacteroidales* orders [11]. Nevertheless, the impact on host microbiota is unknown for most antiparasitic treatments. Regarding the relative abundance of OTUs in the GI of the MNP-treated animals here studied, there was a significant decrease of *Mycoplasmataceae* in abomasum; this family represents 8.85% of the abomasum microbial content of all the animals here studied. The relative abundance of this family was significantly reduced, being 16 times lower than in control animals. *Mycoplasmataceae* is characterised by the lack of a cell wall; thus, mycoplasmas are very small and flexible, and can penetrate the host cells. The predominant clinical presentations of mycoplasmosis in animals encompass respiratory disorders, mastitis, reproductive dysfunctions, arthritis, anaemia and other chronic health issues [37]. A limitation of our experimental design is that we cannot discern whether the observed change in the relative abundance of *Mycoplasmataceae* is a direct effect of the antiparasitic treatment or of the absence of parasites. However, taking into account that there is no direct evidence that MNP may have some activity against mycoplasma, we can hypothesize that the lack of parasites may prevent abomasum colonization by microbes of the *Mycoplasmataceae* family. In this sense, Paz et al., (2022) [38] reported that the whole GI tract of sheep with low epg counts had lower *Mycoplasma* abundance than sheep with high epg counts. The use of a limited number of animals is a constraint of the experimental design used in this

research. Increasing the sample size might allow us to detect differences among OTUs that a smaller sample size might not be able to detect. Undoubtedly, further research is needed to better understand if there is a bacteria-nematode interaction.

Unlike *Mycoplasmataceae*, the family *Methanobacteriaceae*, whose relative abundance in sheep abomasum was 1.89, was increased in 3 out of 5 dewormed animals. Considering that methanobacteria are usually identified in the rumen but not in the abomasum of sheep [38], we believe that this family isolated from the abomasum would actually come down from the rumen of treated sheep. Methanobacteria can pass from the rumen to the abomasum in sheep through a narrow passage called the reticulum-omasal orifice; this passage is normally closed, but it can open to allow food and fluid to pass from the rumen to the abomasum [39]. Our results agree with those of El-Ashram et al. (2017) [40], who found that artificial infection with *H. contortus* decreased drastically the relative abundance of *Archaebacteria* in both rumen and abomasum of 3-month-old sheep. A limitation of the present work is that no samples from ruminal content were evaluated, considering the absence of significant nematode infections in this first ruminant pre-stomach. *Paramphistome* is the only rumen parasite that primarily infects cattle and sheep. Most livestock have only mild stomach fluke infections and generally do not exhibit clinical signs of disease, since they develop resistance after exposure to this parasite [41]. Further studies involving MNP effects on the rumen microbiota as well as measurements of methane production are necessary to define if these differences in *Methanobacteriaceae* abundance are a consequence of the MNP treatment *per se*.

The present work included the use of OTC-LA to induce an imbalance of the sheep GI microbial community and to examine how these changes may affect GI nematode burdens. This drug is a broad-spectrum antibiotic commonly used in veterinary medicine to treat a wide range of bacterial infections in animals, including infections caused by Gram-positive and Gram-negative bacteria, as well as some intracellular bacteria. In this work, 17 days after a single injection of the therapeutic dose of OTC-LA, the gut microbiota of treated animals returned to the control levels, suggesting strong resilience to antibiotic perturbation in the bacterial community of the animals, under the current experimental conditions. The capacity of the microbial community to persist in equilibrium or return to the original state after being disturbed by factors such as diet, stress, and disease, is recognized as the ecosystem resilience [42]. This capacity is key to maintaining the structural and functional stability of the gut microbiota over time [43]. Resilience after microbial manipulation has been largely documented in the literature. In this sense, Dill-McFarland et al. (2019) [44] reported that diet alteration of dairy cattle microbiota may improve production, but changes are often unstable and fail to persist. Holman et al. (2019) [45] demonstrated that the faecal and nasopharyngeal microbiota of cattle is significantly altered after only 5 days of a single injection of OTC or tulathromycin in feedlot systems, with the faecal microbiota being more resilient to antibiotic treatment than the nasopharyngeal one. Human microbiota were found to be resilient and to recover rapidly during antibiotic administration [46]. In this work, a dose of OTC-LA did not produce changes in parasite communities either in the abomasum or in the large intestine between both control and OTC-LA-experimental groups. This result suggests that although the administration of this broad-spectrum antimicrobial causes transient changes in the intestinal microbiota, they are not sufficient to alter the abundance of nematode burden in the GI tract of sheep.

To our knowledge, this is the first work reporting that the MNP-mediated elimination of parasitic GI nematodes induced modifications in the relative abundance of mycoplasmas and methanobacteria. Conversely, extensive manipulation of the GI microbiota through the application of a broad-spectrum antibiotic did not result in significant changes in the GI nematode burden. Furthermore, the bacterial community was able to recolonize and restore the

microbiome balance at 17 days post-antibiotic treatment. Assessing how bacterial and parasite populations may interact in the GI tract may substantially contribute to the understanding of the role of microbiome in nematode parasite establishment and the role of GI parasites in microbiome communities. Overall, this is a first *in vivo* assessment of microbial-parasite interaction in the complex GI tract environment of ruminant species. Microbiome-targeting and other bacteria-based interventions are promising alternative strategies for parasite control in veterinary species and deserve further research.

## Supporting information

**S1 Table. Results of microbiome 16S amplicon metagenomic sequencing.** Base calling accuracy was measured by the Phred quality score, DADA2 denoise and quality filters.
(XLSX)

**S2 Table. Abundance of bacterial taxa in the abomasum.** DESeq2-normalized microbial taxa abundances are shown at the phylum, class, order, and family levels. Alpha diversity indices are indicated.
(XLSX)

**S3 Table. Abundance of bacterial taxa in the large intestine.** DESeq2-normalized microbial taxa abundances are shown at the phylum, class, order, and family levels. Alpha diversity indices are indicated.
(XLSX)

## Acknowledgments

Patricia Cardozo (INTA Balcarce), Paula Dominguez and Lucila Moriones (CIVETAN) collaborated with animal management. We thank translator Jorgelina Brasca for revising the English style.

## Author Contributions

**Conceptualization:** Luis Ignacio Alvarez, Fernanda Imperiale, Carlos Edmundo Lanusse, Juan Pedro Liron.

**Data curation:** Laura Mate, Fernanda Imperiale, Juan Pedro Liron.

**Formal analysis:** Laura Mate, Luis Ignacio Alvarez, Mercedes Lloberas, Fernanda Imperiale, Carlos Edmundo Lanusse, Juan Pedro Liron.

**Funding acquisition:** Laura Mate, Luis Ignacio Alvarez, Carlos Edmundo Lanusse, Juan Pedro Liron.

**Investigation:** Laura Mate, Luis Ignacio Alvarez, Mercedes Lloberas, Fernanda Imperiale, Carlos Edmundo Lanusse, Juan Pedro Liron.

**Methodology:** Laura Mate, Luis Ignacio Alvarez, Mercedes Lloberas, Fernanda Imperiale, Carlos Edmundo Lanusse, Juan Pedro Liron.

**Project administration:** Juan Pedro Liron.

**Resources:** Laura Mate, Luis Ignacio Alvarez, Mercedes Lloberas, Carlos Edmundo Lanusse, Juan Pedro Liron.

**Software:** Laura Mate, Juan Pedro Liron.

**Supervision:** Luis Ignacio Alvarez, Mercedes Lloberas, Juan Pedro Liron.

**Validation:** Laura Mate, Luis Ignacio Alvarez, Mercedes Lloberas, Carlos Edmundo Lanusse, Juan Pedro Liron.

**Visualization:** Laura Mate, Juan Pedro Liron.

**Writing – original draft:** Laura Mate, Juan Pedro Liron.

**Writing – review & editing:** Laura Mate, Luis Ignacio Alvarez, Mercedes Lloberas, Fernanda Imperiale, Carlos Edmundo Lanusse, Juan Pedro Liron.

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
