## [Decision Letter · Decision Letter 0]

22 Mar 2024

PONE-D-23-42901Interaction between bacterial microbiota and nematode parasite communities in sheep’s gastrointestinal tractPLOS ONE

Dear Dr. Liron,

Thank you for submitting your manuscript to PLOS ONE. After careful consideration, we feel that it has merit but does not fully meet PLOS ONE’s publication criteria as it currently stands. Therefore, we invite you to submit a revised version of the manuscript that addresses the points raised during the review process.

**ACADEMIC EDITOR: **

Please carefully address the comments of the reviewers.

We look forward to receiving your revised manuscript.

Kind regards,

Harvie P. Portugaliza, D.V.M., Ph.D.

Academic Editor

PLOS ONE

“This work was supported by CONICET (PIP 11220100102665CO) and INTA, all from Argentina. The authors appreciate the collaboration of Patricia Cardozo (INTA Balcarce), Paula Dominguez and Lucila Moriones (CIVETAN) on the animal management of the reported work.”

“This work was supported by CONICET PIP 11220100102665CO and  the  Agencia Nacional de Promoción Científica y

Tecnológica PICT 2020-0343.”

Additional Editor Comments:

Please carefully address the comments of the reviewers.

Reviewers' comments:

Reviewer's Responses to Questions

**Comments to the Author**

1. Is the manuscript technically sound, and do the data support the conclusions?

Reviewer #1: Partly

Reviewer #2: Yes

2. Has the statistical analysis been performed appropriately and rigorously? 

Reviewer #1: No

Reviewer #2: Yes

3. Have the authors made all data underlying the findings in their manuscript fully available?

Reviewer #1: Yes

Reviewer #2: Yes

4. Is the manuscript presented in an intelligible fashion and written in standard English?

Reviewer #1: Yes

Reviewer #2: No

5. Review Comments to the Author

Reviewer #1: This manuscript from Mate et al describes changes in the microbial gut composition of sheep infected with gastrointestinal nematodes and treated with antibacterial and anthelmintic drug compounds.

It is widely accepted that the microbiome has a large role to play in gut health and that fluctuations in microbial composition can have detrimental impacts to the host. Moreover, there is increasing interest in understanding the role of the microbiome during helminth infection in the gut and how parasite infection might induce changes in microbial community composition and function. In ruminants, this is particularly pertinent given the pressures on producers to maximize feed conversion, as well as the role of the microbial community in methane production and by extension the environment. Parasitic infection could therefore have negative implications for each of these.

In this study, fifteen naturally infected lambs were used. Animals were kept in paddocks during the trial and shared a mixed grass paddock feed, as well as unrestricted access to water. Animals were divided into 5 animals per group. One group received anthelmintic treatment, one group received anti-bacterial treatment and another group was untreated. The authors then analyzed the microbial communities of these animals to determine changes in microbial composition in response to treatment and infection status (since anthelmintic treatment eradicated most worm infections). Abomasum, small intestine and large intestinal samples were collected at post-mortem for microbial and parasitological analyses.

The authors found that oxytetracycline (antibacterial) treatment has not significant impact on the nemabiome composition (shown in figure 1 and table 1).

Both treatments were found to be associated with particular increases and decreases in bacterial OTUs, however power analysis assessment is required for these interpretations to be reliably made.

A drawback to the study is the lack of parasitological analysis of species composition in all animals at the start of the study (prior to any treatment), in order to determine how much the parasite profile might have changed in response to treatment. Another drawback is that the lack of an uninfected control group makes it difficult to interpret the extent to which helminth infection changed microbial community composition at baseline across the treatment groups. Despite these drawbacks, however, the study results are worthy of reporting. But limitations such as these should be clearly stated in the discussion, particularly for facilitating improvements to future study designs where possible.

Major issues:

1) State study limitations highlighted above in the discussion section.

2) The OTU data for abomasum, small and large intestine which is currently only reported in supplemental tables should also be presented similar to the parasitological data (figure 1), showing the proportions of Firmicutes, Bacteroidota, etc as proportional bars and showing this for each individual animal as in figure 1, but for all groups (included as main body figures).

3) The authors report decreases and increases of Mycoplasmatacea and Methanobacteriaceae, respectively, in monipantel-treated sheep compared to untreated sheep. However, 3 MNP animals and 2 control animals had similar Mycoplasmatacea counts, while 2 MNP animals and 4 control animals had similar Methanobacteriaceae counts. Since groups only have 5 animals in them, these numbers consist of a relatively large proportion of each group. Therefore, it is essential that the authors perform power analysis on these data to ensure they are not underpowered. If the data is underpowered, then concluding remarks based on statistical tests should not be made, as it would not be clear if there really is group-dependent increases and decreases in specific bacteria types.

4) In line with the point above (and a similar point that can be made for the OCT group), were the animals that are more different to the control (i.e. where there is greater separation in datapoints) associated with the animals that had persistent Oesophagostomum and Trichuris infection in the large intestine?

5) Please make clearer why there is no section on the small intestine in the microbiome section of the results (only abomasum and large intestine).

Minor issues:

1) Line 189: MNP-treated animals are not shown on figure 1. However, the text at line 189 indicates they are.

2) Figure 1: State which colour is T. axei and which is T. colubriformis in the figure legend (both labelled Trich.sp at present).

3) Figure 4 could be more readily interpreted if each panel (A-D) had a title (e.g. abomasum, large intestine, etc).

Reviewer #2: The present work describes how an antibiotic treatment with oxytetracycline or an anthelmintic, monepantel, may affect the microbiota of the gastrointestinal tract of sheep naturally infected by gastrointestinal nematodes.

Although the subject is of great interest, and more studies are needed to determine the parasite-microbiota interaction, in the present work there are several aspects that are not clearly explained, mainly in relation to the methodology. In addition, and mainly in the discussion, the authors should compare their results with other studies in which the microbiota in sheep infected by gastrointestinal nematodes have been studied. Since in general there are not many studies on this subject, the few that exist should be taken into account.

The following are the points that authors should clarify

Reference 5 is not correct: Kinetics of acute infection with Toxoplasma gondii and histopathological changes in the duodenum of rats. Please, review other papers in which interaction microbiota and gastrointestinal nematode has been studied.

Line 127. It is not clear how many treatments the animals received. It seems that sheep were treated 3 times, at days 0, 7 and 14 days. Is this correct? If so, authors should explain the reason of this protocol.

Line 131. What do authors mean with "17 days post-treatment the animals were euthanized"? again, in the MNP group, when were the animals euthanized? 17 days after the last treatment at day 14?. This is not clear in the text.

Line 133. Were the content samples collected for microbiome analysis? Explain correctly the purpose of the collection of these samples, but also the "parasitological analysis". On the other hand, in the results section, only abomasum and large intestine are included. Were the samples from small intestine analysed?.

Line 139. And for the content samples, how were they stored?. Which was the aim of this DNA extraction in feacal samples?. In the results section these samples are not included.

In the “Animals and experimental design” section, authors should add when the feacal samples were collected for coproculture and, in which groups.

Line 149. This sentence is not explained correctly. If the deep amplicon sequencing was done with L3, it is incorrect to say that the genus of parasites recovered in the intestine will be assigned according to these results. Please, explain this.

Line 150. How were the adults counted? the technique used is not described. And, were they identified?

Line 155. DNA extraction only in feacal samples?. What happed with the abomasum and large intestine content?. The authors are not clear about the samples they worked with.

Results. “Parasitological analysis” section. Did the authors analysis the number of eggs per gram in faeces in all groups? Before and after treatment. If so, please, include the data and the result of the monepantel efficacy. On the other hand, regarding the results, please, explain the results for each treatment, independently, not mixing the results of the different groups.

Line 189. Again, this part of the work is not clear. With which samples was the ITS amplicon sequencing performed? L3?. Did the authors identify the adults collected by morphological techniques?

Regarding the parasitological analysis, please, explain the results for each treatment, independently, not mixing the results of the different groups.

Line 236. MNP group. Revise and standardize group names throughout the text.

Line 305. Authors are mixing the identification in L3 and adults, and this is not correct.

Line 309. Please, do not mix the results of both groups. Explain the results for each group separately.

Discussion. Improve the discussion comparing the results of the microbiota with other studies with sheep infected with GIN.

Table 1. How was the identification of adult parasite species performed?. Adult parasite counts....

Figure 1. Are these results from L3? Please, clarify it in the text.

6. PLOS authors have the option to publish the peer review history of their article (what does this mean?). If published, this will include your full peer review and any attached files.

Reviewer #1: No

Reviewer #2: No

---

## [Author Response · Author response to Decision Letter 0]

3 May 2024

Dear reviewers,

Thank you very much for the review of this manuscript. The comments have certainly helped us a lot to improve the content of our work and we think that this version is noticeably better than the previous one. Changes in the MS were highlighted in yellow. English style was revising by a translator. Please find below the answers to each of your questions and comments. 

Yours sincerely,

Juan Pedro Liron

Reviewer #1: 

A drawback to the study is the lack of parasitological analysis of species composition in all animals at the start of the study (prior to any treatment), in order to determine how much the parasite profile might have changed in response to treatment. 

We acknowledge the reviewer's suggestion regarding a potential experimental design involving pre- and post-treatment comparisons for each experimental group. However, it is not possible because the determination of the parasitological content in the gastrointestinal tract involves the sacrifice of the animals. To avoid killing four groups of animals, our experimental design involved dividing the experimental lambs into three groups (control, MNP and OTC), each consisting of five lambs that were gender- and parasitic burden-balanced. Post sacrifice recovered parasites from abomasum, small and large intestine content were counted through the methodology applied to perform the controlled efficacy test (CET) and the comparison was MNP vs Control and OTC-LA vs Control group.

Another drawback is that the lack of an uninfected control group makes it difficult to interpret the extent to which helminth infection changed microbial community composition at baseline across the treatment groups. Despite these drawbacks, however, the study results are worthy of reporting. But limitations such as these should be clearly stated in the discussion, particularly for facilitating improvements to future study designs where possible.

As the reviewer #1 suggest, the inclusion of a group of uninfected animals would have been “ideal” to determine how helminth infection changed microbial community composition at baseline across the treatment groups. However, under extensive farming conditions, it is impossible to find natural uninfected lambs. We recreated this condition by deworming the animals (MNP group). The limitation of this experimental design is that we cannot discern whether the observed changes in bacterial communities are a direct effect of the antiparasitic treatment or the absence of parasites. This limitation was included in the discussion of the MS. 

Major issues:

1) State study limitations highlighted above in the discussion section.

Thanks for the comment. We have revised the discussion section accordingly.

2) The OTU data for abomasum, small and large intestine which is currently only reported in supplemental tables should also be presented similar to the parasitological data (figure 1), showing the proportions of Firmicutes, Bacteroidota, etc as proportional bars and showing this for each individual animal as in figure 1, but for all groups (included as main body figures).

The following figures (see the attach file response to reviewers) illustrate the relative abundance of OTUs in the abomasum and large intestine of each animal, respectively. These Figures were included in S2 and S3 Tables. We are open to the suggestion of the reviewer and/or editor to include these figures in the MS if they deem it appropriate.

Abomasum Large Intestine

3) The authors report decreases and increases of Mycoplasmatacea and Methanobacteriaceae, respectively, in monipantel-treated sheep compared to untreated sheep. However, 3 MNP animals and 2 control animals had similar Mycoplasmatacea counts, while 2 MNP animals and 4 control animals had similar Methanobacteriaceae counts. Since groups only have 5 animals in them, these numbers consist of a relatively large proportion of each group. Therefore, it is essential that the authors perform power analysis on these data to ensure they are not underpowered. If the data is underpowered, then concluding remarks based on statistical tests should not be made, as it would not be clear if there really are group-dependent increases and decreases in specific bacteria types.

We appreciate the reviewer's valuable feedback. The Wald test performed by Dseq2 is a widely used and robust statistical test that takes into account the variability of the data. Surely, increasing the sample size in an essay can enhance the sensitivity of the test, allowing for the detection of differences between OTUs that a smaller sample size might not be able to detect. However, this would imply the sacrifice of a greater number of animals. Animal ethics and welfare are a priority for us. Minimizing the number of animals used in research is fundamental to our work. We have included in the discussion a paragraph about the limitation of the animal’s number, a consequence of the experimental design applied in this study.

4) In line with the point above (and a similar point that can be made for the OCT group), were the animals that are more different to the control (i.e. where there is greater separation in datapoints) associated with the animals that had persistent Oesophagostomum and Trichuris infection in the large intestine?

The number of Oesophagostomum and Trichuris recovered from large intestine content was too small (only 1 parasite in the 1% of the whole intestinal content) to be considered biologically and statistically significant. Similarly, only one animal (JPL6A) from MNP group had a very low count of Oesophagostomum and another different animal (JPL8A) from this group had a very low count of Trichuris. It was similar for the animals from OTC-LA group.

 5) Please make clearer why there is no section on the small intestine in the microbiome section of the results (only abomasum and large intestine).

We acknowledge the reviewer's comment, we corrected the manuscript because it was unclear where the samples for bacterial analysis were taken from. After MNP and OTC-LA treatment, we waited for microbiota stabilization and 17 days post treatment, animals were euthanized and samples from abomasum, small and large intestine content were taken for microbiome analyses. DNA from abomasum and large intestine with the quality and amount required for metagenomic analyses was obtained. By contrast, after two unsuccessful attempts, only in 6 out 15 small intestine samples we were able to obtain the DNA required for metagenomic analyses. For this reason, small intestine was not included for metagenomic sequencing. This explanation was included in the results section (lines 207-211). 

Minor issues:

1) Line 189: MNP-treated animals are not shown on figure 1. However, the text at line 189 indicates they are.

Thanks for the observation, the correction was made in the MS.

2) Figure 1: State which colour is T. axei and which is T. colubriformis in the figure legend (both labelled Trich.sp at present).

Figure 1 was modified accordingly.

3) Figure 4 could be more readily interpreted if each panel (A-D) had a title (e.g. abomasum, large intestine, etc).

Thanks for the observation, the titles were included in the Figure 4.

Reviewer #2: The following are the points that authors should clarify

Reference 5 is not correct: Kinetics of acute infection with Toxoplasma gondii and histopathological changes in the duodenum of rats. Please, review other papers in which interaction microbiota and gastrointestinal nematode has been studied.

We have replaced Trevizan et al. 2016 reference.

Line 127. It is not clear how many treatments the animals received. It seems that sheep were treated 3 times, at days 0, 7 and 14 days. Is this correct? If so, authors should explain the reason of this protocol. 

Line 131. What do authors mean with "17 days post-treatment the animals were euthanized"? again, in the MNP group, when were the animals euthanized? 17 days after the last treatment at day 14?. This is not clear in the text.

Line 133. Were the content samples collected for microbiome analysis? Explain correctly the purpose of the collection of these samples, but also the "parasitological analysis". On the other hand, in the results section, only abomasum and large intestine are included. Were the samples from small intestine analysed?.

Line 139. And for the content samples, how were they stored?. Which was the aim of this DNA extraction in feacal samples?. In the results section these samples are not included. In the “Animals and experimental design” section, authors should add when the feacal samples were collected for coproculture and, in which groups.

We acknowledge the reviewer's comments (Lines 127, 131, 133, 137 and 139). We re-wrote the animals and experimental design section (lines 115-132) because it was unclear where, when and why the samples for bacterial and parasitological analyses were taken.

Line 149. This sentence is not explained correctly. If the deep amplicon sequencing was done with L3, it is incorrect to say that the genus of parasites recovered in the intestine will be assigned according to these results. Please, explain this.

This sentence was clarified in the MS. The morphological identification of adult parasites recovered from each intestinal segment reach the genus level. ITS-2 deep amplicon sequencing (nemabiome analyses) of L3 allows us to know the species of each genus infecting the animals. 

Line 150. How were the adults counted? the technique used is not described. And, were they identified?

Adult parasites from abomasum, small and large intestine content were counted and identified through the methodology applied to perform the controlled efficacy test (CET), following the World Association for the Advancement of Veterinary Parasitology (WAAVP) guidelines [Coles et al. 1992]. Please see parasitological analyses section. This technique was not described in the MS because it is the gold standard technique used and recommended in veterinary parasitology sciences.

Line 155. DNA extraction only in feacal samples?. What happed with the abomasum and large intestine content?. The authors are not clear about the samples they worked with.

We corrected the animals and experimental design section of the MS because it was unclear where the samples for bacterial analyses were taken from. Intestinal content from the different intestinal segments were taken for adult parasite identification and microbiome analyses. Faecal samples were used only for coproculture in order to obtain L3 for nemabiome analyses. 

Results. “Parasitological analysis” section. Did the authors analysis the number of eggs per gram in faeces in all groups? Before and after treatment. If so, please, include the data and the result of the monepantel efficacy. On the other hand, regarding the results, please, explain the results for each treatment, independently, not mixing the results of the different groups.

At day 0 faecal samples were taken from about 30 lambs located in a paddock on an experimental sheep farm (Reserva 8, Instituto Nacional de Tecnología Agropecuaria, Balcarce, Argentina) in order to proceed with the faecal egg count. Fifty out of 30 lambs, gender- and parasitic burden-balanced, were assigned to three experimental groups (Control, MNP and OTC-LA) of five lambs each. Animals in MNP and OTC-LA received the corresponding treatment. To perform parasitological analyses intestinal content from abomasum, small and large intestine was collected after animal sacrifice at day 17. Adult parasites were morphological identified (genus level), for all the animals from the three groups. Animals belonging to MNP group did not have gastrointestinal parasites (with the exception of a very low count of Oesophagostomum and Trichuris) and for this reason this group was not included in the Table 1. Pools of faecal samples from each experimental group were taken before animal sacrifice for coproculture, in order to performed nemabiome analyses. This technique allows us to identify the nematode species infecting each experimental group. Table and Figure 1 depicted species parasite information. 

Line 189. Again, this part of the work is not clear. With which samples was the ITS amplicon sequencing performed? L3?. Did the authors identify the adults collected by morphological techniques?

Regarding the parasitological analysis, please, explain the results for each treatment, independently, not mixing the results of the different groups.

We have re-write the parasitological analyses (Results section). 

Line 236. MNP group. Revise and standardize group names throughout the text. 

Group names were revised and corrected in the MS, Tables and Figures.

Line 305. Authors are mixing the identification in L3 and adults, and this is not correct.

This sentence was deleted from the discussion.

Line 309. Please, do not mix the results of both groups. Explain the results for each group separately.

This sentence was clarified.

Discussion. Improve the discussion comparing the results of the microbiota with other studies with sheep infected with GIN.

We appreciate the suggestion. While works focused on the interaction between naturally occurring gastrointestinal parasites in sheep and the host microbiome is scarce, we have improved the discussion by comparing our findings to relevant existing research, even if those studies involved artificial infections with only one parasite genus.

Table 1. How was the identification of adult parasite species performed?. Adult parasite counts.... Figure 1. Are these results from L3? Please, clarify it in the text.

It was clarified above.

---

## [Decision Letter · Decision Letter 1]

18 Jun 2024

Interaction between bacterial microbiota and nematode parasite communities in sheep’s gastrointestinal tract

PONE-D-23-42901R1

Dear Dr. Liron,

We’re pleased to inform you that your manuscript has been judged scientifically suitable for publication and will be formally accepted for publication once it meets all outstanding technical requirements.

Kind regards,

Harvie P. Portugaliza, D.V.M., Ph.D.

Academic Editor

PLOS ONE

Additional Editor Comments (optional):

Reviewers' comments:

Reviewer's Responses to Questions

**Comments to the Author**

1. If the authors have adequately addressed your comments raised in a previous round of review and you feel that this manuscript is now acceptable for publication, you may indicate that here to bypass the “Comments to the Author” section, enter your conflict of interest statement in the “Confidential to Editor” section, and submit your "Accept" recommendation.

Reviewer #2: All comments have been addressed

2. Is the manuscript technically sound, and do the data support the conclusions?

Reviewer #2: Yes

3. Has the statistical analysis been performed appropriately and rigorously? 

Reviewer #2: Yes

4. Have the authors made all data underlying the findings in their manuscript fully available?

Reviewer #2: Yes

5. Is the manuscript presented in an intelligible fashion and written in standard English?

Reviewer #2: Yes

6. Review Comments to the Author

Reviewer #2: All comments have been addressed. However, the authors could have included other publications to compare the results with those of other studies.

7. PLOS authors have the option to publish the peer review history of their article (what does this mean?). If published, this will include your full peer review and any attached files.

Reviewer #2: No

---

## [Editor Report · Acceptance letter]

20 Jun 2024

PONE-D-23-42901R1 

PLOS ONE

Dear Dr. Liron, 

I'm pleased to inform you that your manuscript has been deemed suitable for publication in PLOS ONE. Congratulations! Your manuscript is now being handed over to our production team.

Kind regards, 

on behalf of

Dr. Harvie P. Portugaliza 

Academic Editor

PLOS ONE